# Non-Dispersive Anti-Washout Grout Design Based on Geotechnical Experimentation for Application in Subsidence-Prone Underwater Karstic Formations

**DOI:** 10.3390/ma14071587

**Published:** 2021-03-24

**Authors:** Khaqan Baluch, Sher Q Baluch, Hyung-Sik Yang, Jung-Gyu Kim, Jong-Gwan Kim, Saeed Qaisrani

**Affiliations:** 1Department of Energy and Resources Engineering, Graduate School, Chonnam National University, Gwangju 61186, Korea; hsyang@jnu.ac.kr (H.-S.Y.); evangelong@hanmail.net (J.-G.K.); 2Consulting Engineer, EDACS International, 6000 Ohrid, North Macedonia; Sherbaluch@hotmail.com; 3Department of Environmental Sciences, COMSATS University, Islamabad 61000, Pakistan; Saeed.qaisrani@cuivehari.edu.pk

**Keywords:** anti-washout and non-dispersive cement grouts, underwater construction, anti-washout admixtures, multi-factor optimization

## Abstract

A new non-dispersive, anti-washout grout consisting of ordinary Portland cement, slag, superplasticizer, and methylbenzyl cellulose is proposed herein for the treatment of open karst, jointed and fractured rock, open-work gravel, and permeable sediments. A series of laboratory experiments were performed to design an anti-wash out grout suitable for grout injection of coarse aggregates depicting partially and open-jointed saturated rock mass and grouting concrete aggregates for underwater construction. The Taguchi orthogonal array was used to obtain nine different grout mix ratios. A total of four variables were considered, each with three different levels of the water–cement ratio, slag, and dosage of additives such as the superplasticizer and methyl benzyl cellulose. The laboratory determination of grout characteristics recording of mini slump, temperature, pH, visual assessment of grout dispersion, bleeding, and initial setting time and as well as uniaxial compressive strengths and permeabilities of the hardened grout samples were tested. To evaluate the suitability of the grout mixes, an analysis of variance was used for factor analysis and Grey relational analysis (GRA) was used to determine the optimal grout mix design. Based on the GRA, the following levels of the factors afforded the best results: water level 1 (0.3%), SP level 3 (0.01%), methylbenzyl cellulose level 2 (0.002%), and slag level 3 (0.1%). This paper describes the research methodology, detailed research observations, and analyses involved in designing the appropriate concrete mix. Based on the conclusions, relevant commendations regarding the suitability of grout testing equipment and grout mix designs are presented.

## 1. Introduction

The present research deals with the investigation of formation of sinkholes near Jangseong limestone mine. However, the limestone formation, as evidenced in the mine providing raw material for manufacture of cement, is already karstic, containing partially infilled cavities and open voids in limestone.

As result of phased investigations comprised of satellite imagery, mapping of sinkholes, borehole drilling, and piezometric observations, it became evident that sinkholes are largely formed by aggressive groundwater movement dissolving limestone as it percolates along joints and fractures. Relatively high gradient in the groundwater table reflects significant water flow towards the nearby river. A numerical analysis modelling multiple level of limestone extraction in the mine indicated that the sinkholes are not directly related to the mining activity. Investigations into previous grouting treatment in the mine to reduce water inflow showed that the normal OPC cement grout was not effective, because this type of grout has long setting and hardening time and is also prone to dispersion and erosion.

Grouting is a geotechnical process that involves injecting a chemical grout to fill cracks or voids in the rock mass or coarse-grained soil. Cement is the most common and major ingredient in grouts. It is necessary to understand the rock conditions and properties before initiating grouting [1,2]. Both the soil formation and grout characteristics should be considered when choosing the grout. Therefore, permeability testing of the rock mass or soil must be performed. Furthermore, the rock cores and particle size distribution of soils must be analyzed. For long-term durability, the grout must have low sedimentation, resistance to wash out, good flow properties, early strength gain, and compatibility with the surrounding environment [3]. The most commonly used grout consists of cement and water, along with additives that improve workability and applicability. When the voids are large and easy to penetrate, fillers are used to increase the bulk of the grout mix. The filler properties also influence grout strength. Generally, grout strength is not a major concern provided the grout is not erodible and its strength is not significantly low (<8~12 MPa). Sand is a cheap filler but requires care to avoid segregation. Clays, including bentonite, can be used as grout fillers or as a grout on their own; however, clays are costly and require careful mixing and agitation prior to injection in order to maintain workability and have low compressive strength upon hardening.

A detailed literature review was carried out to determine the state of art development of non-dispersive grouts for use in void and cavity filling and use in underwater construction. Based on the previous examples and case histories, a series of laboratory grouting trials were performed to meet the non-dispersive grouts’ requirements.

The main grout parameters were selected using Taguchi orthogonal array to obtain nine different grout mix ratios. A total of four variables were considered, each with three different levels of the water–cement ratio, slag, and dosage of additives such as the superplasticizer and methyl benzyl cellulose. The laboratory determination of grout characteristics were based on recording of mini slump, temperature, pH, visual assessment of grout dispersion, bleeding, and initial setting time. Uniaxial compressive strengths and permeabilities of the hardened grout samples were evaluated as well. To evaluate the suitability of the grout mixes, an analysis of variance was used for factor analysis and Grey relational analysis (GRA) was used to determine the optimal grout mix design. Based on the GRA, the following levels of the factors afforded the best results: water level 1 (0.3%), SP level 3 (0.01%), methylbenzyl cellulose level 2 (0.002%), and slag level 3 (0.1%).

The laboratory testing of non-dispersive grout mixes led to an optimized, stable, non-dispersive and non-erodible grout mix with adequate workability and compressive strength requirements, which is proposed for field trials and use in underwater construction, filling of partially filled cavities and open voids, and treatment of jointed rock.

## 2. Literature Review

Li developed a new cementitious anti-washout grouting material (CIS), Cementitious grouts for Instant Sealing, which was prepared using ordinary Portland cement, water glass, and xanthan gum. The CIS grout offers the advantages of a short setting time, high early mechanical strength, high slurry viscosity, high slurry retention rate, and nontoxicity; the highest compressive strength achievable is 5 MPa. However, because of the low slump spread and setting time, long-distance pumping is not possible [4]. Jeff prepared anti-washout admixtures based on water-soluble polymers of various types for use in concrete placed underwater to minimize cement washout and ensure in-place concrete of satisfactory quality [5]. Cui made two anti-washout grouts using Portland cement, accelerating agents, and flocculating agents (cationic polyacrylamide, hydroxyethyl cellulose ether, underwater binder-II, and bentonite); two unnamed accelerating agents were also used. Two different grouts with a maximum slump spread of 60 mm were technically pumpable, featuring a maximum 28-day compressive strength of 13 MPa. This grout was developed to control water flow [6]. Various materials and admixtures have been introduced and used to control segregation and dispersion of cementitious materials. For instance, lignosulfonate was made in the 1920s and used as a concrete plasticizer biopolymer; lignite was established in the 1940s and used as a bentonite thinner biopolymer; Xanthan gum was introduced in the 1960s as a viscosity modifier biopolymer; melamine and naphthalene condensates were developed in 1962 and used as concrete superplasticizer synthetic polymers; cellulose ethers were introduced in the 1970s as water-retention agent biopolymers; vinylsulfonate copolymers were suggested in the 1980s and used as water-retention agents synthetic polymers; polycarboxylate copolymers were also put forward in the 1980s and used as concrete superplasticizer synthetic polymers; and polyaspartic acid was originated in the 1990s and used as a retarder biopolymer to increase the setting time of concretes and cementitious grouts [7]. Twenty different types of cellulose have been developed and used in various applications. In the construction field, cellulose has been used by various researchers for water retention, flowability, workability, strength, and non-dispersity. Hideo Tawara proposed an underwater, anti-washout, non-shrink grout using hydroxypropyl methylbenzyl cellulose [8]. Changmin initiated an anti-washout concrete mix for underwater use using methylbenzyl cellulose gum [9]. Bray tested cement additives for sealing gas channels and water wells [10]. Sahara used aggregates and methylbenzyl cellulose to reduce the flow of concrete [11]. In the present research, a wide-ranging DOE (design of experiments) study was adopted to develop a new type of cement-based, anti-washout grout. Tests were performed to study the effects of various constituents on the mini slump spread, permeability, compressive strength, pH, dispersion, and temperature. Grey relational analysis and the Taguchi design were used to obtain a simple and cost-effective method for the multiparametric optimization of cement grout mixes.

## 3. Grout Materials and Research Methodology

A new non-dispersive anti-washout grout consisting of ordinary Portland cement, slag, superplasticizer (Dongnam Flowmix 3000S, Bongam-dong, Korea), and VMA (methylbenzyl cellulose) is prepared. This grout is different from existing admixtures and viscosity modifiers such as micro-silica, nano-silica slurry, high molecular ethyl-enoxy derivate, natural polysaccaride, methyl cellulose, methyl propyl cellulose, and starch derivate. It is also different from other materials such as various types of accelerators, xanthan gum, clay, and HI-FA (High Performance and Multi-Functional Agent) materials. Blast furnace slag (BFS) is a calcium silicate based by-product produced from melted iron ore in the blast furnace. Typically, it is swiftly cooled to a glassy state and it is pulverized into various grades of fineness for the use as filler in the cementitious construction materials. BSF is widely used between 15–35% of cement content to control heat of hydration and for improvement in workability of cement without impairing its strength compressive strength. BSF has dormant hydraulic properties that allows its most communal application for being a cement additive and in concrete structures. This property allows its part in soil stabilization and in mortar for masonry. The chemical components of BSF are given in Table 1 and particle size distribution is given in Figure 1.

### 3.1. Selection of Target Permeability and Aggregates Size

The selection of the size of the aggregate was based on the permeabilities of jointed and open-jointed rocks. The permeabilities were 10^−3^ and 10^−2^ m/s, respectively. Therefore, 16-mm and 10-mm aggregates were selected for the open and jointed rocks, respectively. As depicted in Figure 2, sieve analysis was performed on 50% of each aggregate to confirm the average size of each aggregate. The results are listed in Table 2 and Table 3. After the sieve analysis, each aggregate was put into a model, as depicted in Figure 3.

A falling head permeability test was performed, and permeability was determined using Equation (1). The permeabilities of the 16-mm and 10-mm aggregates were 10^−2^ and 10^−3^ m/s, respectively. It is necessary to determine permeability considering the actual rock/soil condition to ensure that aggregates with the appropriate permeability are used to design the most suitable grout in consideration of the actual conditions. After the target permeability was obtained, nine different grouts were designed and implemented using two types of aggregates in an underwater environment.
(1)$$k=2.3\frac{aL}{At}\log(\frac{h1}{h2})$$

In the above-given equation, “*k*” is the coefficient of permeability (m/s), “*a*” is the diameter of the main model pipe (m), “*L*” is the total length of the sample present in the sample section (m), “*A*” is the cross-section of the sample (m), “*t*” is the time in which water drops from *h*_1_ height to *h*_2_ height, *h*_1_ and *h*_2_ are the water initial height and final height, respectively, and the log is base 10 logarithm.

### 3.2. Grout Preparation and Testing

To execute the grout, the following materials were used: superplasticizer, methylbenzyl cellulose-based viscosity modifying admixture, Portland cement, and water. To optimize the mixing ratios of the materials, the Taguchi orthogonal array was designed as in Table 4. Based on the Taguchi orthogonal array, nine grouts were prepared using the proportions listed in Table 5. The 28-day strength of each grout specimen and the target permeability of the aggregate were evaluated in an underwater environment. Subsequently, the mass permeability and strength were assessed. Using these outcomes, Grey relational analysis was used to obtain the most significant variables and optimal levels that improved the grout strength and other properties. The results were validated by directing assenting experiments that show good treaties with optimum design results.

Based on the Taguchi array, all grouts were first prepared and tested in the second phase, before implementing and testing them on the target aggregates. The following properties were checked: pH, temperature, bleeding, dispersion, mini slump spread, initial setting time, and compressive strength. For the pH and temperature, the OHAUS starter3100 (Seoul, Korea) was used immediately after preparation of each grout, as in Figure 4.

For the bleeding test, each grout mix was poured into a graduation cylinder. The bleeding percentage was measured 45 min later, as depicted in Figure 5.

The dispersiveness of the grout samples was checked by pouring the grout into a water-filled beaker, as depicted in Figure 6.

Based on the standard 200 to 300 mm slump spread and the best flowable grout, for the mini slump spread, an open-ended flow cylinder with dimension of 75 mm × 150 mm was used to measure the spread of the grout [12]. The spread of the grout was measured along two directions, both perpendicular to each another, and the average of the spread was noted (see Figure 7).

For the initial setting time, the Vicat apparatus (HD-102, Hyundai, Seoul, Korea) was used (as shown in Figure 8). The grout was tested after 1 h to obtain the results.

Compressive strength is an important factor. To test the compressive strength of the grout mixes, cubes samples were cast in bronze molds with dimensions of 50 × 50 × 50 mm^3^, as depicted in Figure 9. From the next day onward, the cubes were submerged in water for curing for 28 days. Thereafter, the mean strength values of the grouts were noted.

In the second phase of testing, each grout sample was added to the aggregate in the presence of water. Grout penetration was evaluated; after solidification, the sample was placed in water for seven days for curing, because a low 28-day strength was predicted based on ASTM D6103 [13]. Special molds with a diameter of 230 mm and length of 200 mm were prepared. The aggregate was placed into the molds after compaction. The molds were filled with water, and a 65-mm-diameter pipe was inserted through the aggregate to the base of the mold for injecting the grout, as shown in Figure 10.

Two samples of each grout type were prepared: one with the jointed rock permeability fine aggregate (10-mm) and the other with the open-jointed rock permeability coarse aggregate (16-mm). Thus, a total of 18 samples were prepared. After seven days of curing, the samples were dried, and a permeability test was performed. Each sample was placed underwater for one day at a maximum pressure of 1 MPa. Typically, pressure is maintained at 0.5 MPa, but it was increased to 1 MPa to reduce the test time from two days to one. Following a test as per the British standard BSEN 12390 [14], the maximum depth of the water penetration traces was measured. A sample of each grout mix was drilled, and two cylinders of each sample with fine and coarse aggregates were created using a rotary drill. A total of 36 samples were extracted, as depicted in Figure 11.

After the cylinders were prepared, their upper and lower faces were leveled using a precise core cutter, and 36 samples were tested in terms of their compressive strength (DYHU-100TC, Daeyeong Precision, Gunpo-si, Korea), as depicted in Figure 12. All the tests were performed as per the ASTM standard [15]. Each sample had a diameter of 100 mm and length of 200 mm.

## 4. Use of Minitab and GRC Analysis for the Optimization, Results of Lab Experiments

### 4.1. Results of the Lab Experiments

As discussed above, the following tests were performed on the cylinders with the coarse aggregates in the laboratory: mini slump spread (mm), initial setting time (min), dispersion (yes/no), bleeding (%), pH and temperature (°C), uniaxial compressive strength (UCS), for the cubes, and cylinders with fine and coarse aggregate. The results of these tests are listed in Table 6 and Figure 13.

Figure 13 depicts the histograms of mini slump spread, strength of the cubes, mass-strength of the samples with fine and coarse aggregates for nine different grout mixes, and water penetration depth. It revealed that Grout 1 exhibited the lowest mini slump spread, whereas Grouts 8 and 9 exhibited the highest mini slump spreads. The strength of Grout 1 was the highest, whereas Grout 9 had the lowest strength. Grout 1 had the lowest penetration depth of 20 mm for the fine aggregate, as compared with that of the coarse aggregates sample (45 mm). Grout 9 had the highest penetration depths of 130 and 150 mm for the fine and coarse aggregates, respectively.

### 4.2. Use of Minitab and GRA

In this study, to obtain the design matrix, the Taguchi orthogonal array was used together with a restricted number of experiments for the parametric space. The experiments were performed based on the Taguchi orthogonal array design, which is used to optimize engineering problems. However, this process is suitable for single-parameter optimization [16]. Therefore, multiple responses cannot be addressed using the Taguchi method [17]. Consequently, researchers often use GRA with PCA for optimizing multiple responses instantaneously. This technique is different from the single-response optimization. This is an active statistical method and offers relatively effective results in obtaining a blend of parameters for the optimization of multiple responses. The concept of GRA-PCA is depicted in Figure 14.

The signal-to-noise (*S*/*N*) ratio for the larger the better criterion is calculated from Equation (2):(2)$$\frac{s}{N}\mathrm{ratio}=\left(-10\right)\;*\;\log_{10}\left(\frac{1}{x}\right)\overset{x}{\underset{i=1}{\sum }}\frac{1}{y_{ij}^{2}}$$

The *S*/*N* ratio for the smaller-the-better criterion is calculated from Equation (3), where *x* is the number of duplicates, and *y_ij_* is the calculated observation:(3)$$\frac{s}{N}\mathrm{ratio}=\left(-10\right)\;*\;\log_{10}\left(\frac{1}{x}\right)\overset{x}{\underset{i=1}{\sum }}y_{ij}^{2}$$

The grouting process has multiple responses and the grouting quality depend strongly on optimizing all the responses instantaneously. GRA is commonly used by researchers, because statistical techniques can obtain better results for multiple-response optimizations [18]. In this study, the objectives were the maximization of the compressive strength and mini slump; therefore, a larger-the-better criterion is used for these excellence characteristics, and the regularized results are expressed using Equation (4):(4)$$y_{j}^{*}\left(q\right)=\frac{y_{j}\left(q\right)-\min\;y_{j}\left(q\right)}{\max\;y_{j}\left(q\right)-\min\;y_{j}\left(q\right)}$$

However, bleeding, temperature, and initial setting time need to be minimized; to this end, the smaller-the-better criterion is used, as expressed in Equation (5):(5)$$y_{j}^{*}\left(q\right)=\frac{\max\;y_{j}\left(q\right)-\;y_{j}\left(q\right)}{\max\;y_{j}\left(q\right)-\min\;y_{j}\left(q\right)}$$
where *y*_j_(*q*) is the Grey relational value, and max *y_i_*(*q*) and min *y_i_*(*q*) are the largest and smallest values of *y_j_*(*q*) for the observation of the *q*th response, respectively. The number of response variables was 4. The nine annotations of the trials are in sequence *y_i_*(*q*), *j* = 1, 2, …, 9.

A value of 1 is considered as the best normalized result; hence, the value of the normalized results is expected to be larger for accomplishing the best performance.

By calculating the Grey relational coefficients (GRC), data normalization is performed, which describes the relationship between desired and actual experimental normalized results.

For the expression of GRC, ξ*j*(*q*) is determined as shown in Equation (6):(6)$$\varepsilon (y_{j}^{*}\left(q\right),\;y_{j}\left(q\right))=\frac{\Delta_{\min}\left(q\right)-C\;\Delta_{\max}\left(q\right)}{\Delta_{0j\;}\left(q\right)+\;\Delta_{\max}\left(q\right)}$$

The distribution of the experimental data is measured using probability plots, as depicted in Figure 15 and Figure 16. The outlier detection from normality is evaluated using a powerful statistical tool called the Anderson Darling (ADT) test [19]. Figure 15 and Figure 16 depict that the experimental data for all responses are close to the fitted line, the ADT statistics are comparatively low, and the *p*-value of the test is more than 0.05. Hence, it is considered that the data follow a normal distribution. Further analyses and optimization can be performed using these results.

In contrast, Δ_0*i*_(*q*)=│*y*_0_*(*q*) *– y*(*q*) represents the deviation sequence, which is the difference between the reference sequence *y*_0_*(*q*) and the comparability sequence *y*_j_*(*q*). The coefficient (*ᶓ*) takes values such as (0,1), which are set to 0.5 in this analysis [20].

The strength of the correlation between experimental runs is determined by the Grey relational grade (GRG), which is calculated using the mean weights of all the GRGs. (0,1) are the values between these GRG values.

Typically, a larger GRG for an experimental run is ideal, which demonstrates a strong correlation between the ideally normalized value and the corresponding experimental values.

When equal weights are chosen for the quality responses, Equation (7) is used to calculate the GRG [21]:(7)$$\gamma_{j}(y_{0}^{*},y_{j}^{*})=\frac{1}{n}\overset{n}{\underset{q=1}{\sum }}\varepsilon (y_{j}^{*}\left(q\right),y_{0}^{*}\left(q\right))$$

To study the main effects of the input parameters on individual responses, an analysis of variance (ANOVA) was performed at 95% confidence intervals.

For the mini slump, pH, bleeding, UCS of cubes, initial setting time, and compressive strength for the cylinders of coarse and fine aggregates, ANOVA results are listed in Table 7; a *p*-value less than 0.05 shows that the parameter is significant.

The mini slump spread was influenced the most by water (55.74%). When the water content increased or decreased, the mini slump spread also increased or decreased. The other influencing factors were SP (25.09%), VMA (11.65%), and slag (0.58%).

The initial setting time was affected the most by SP (59.95%), followed by water (32.81%), VMA (3.33%), and slag (0.04%). The UCS of the cubes was impacted the most by VMA (43.93%), followed by SP (29.84%), water (24.22%), and slag (0.18%). The significance order for bleeding was water (44.93%), SP (37.13%), VMA (9.28%), and slag (1.54%).

Furthermore, pH was affected the most by slag (38.22%), followed by water (31.51%), SP (15.44%), and VMA (11.34%). With regard to temperature, slag had the most significant impact (50.81%), followed by water (35.42%), VMA (6.38%), and SP (3.43%).

The permeability water penetration depth for the fine aggregate sample was influenced the most by water (55.69%), followed by slag (31.45%), SP (9.77%), and VMA (1.79%). The permeability water penetration depth for the coarse aggregates sample was influenced the most by SP (76.91%), followed by water (15.33%), slag (4.26%), and VMA (0.22%). The cylinder strength of the fine aggregates with lower permeability was influenced the most by water (62.91%), followed by VMA (16.26%), slag (8.80%), and SP (0.79%). Lastly, the cylinder strength of the coarse aggregates was impacted the most by VMA (31.04%), by water (29.84%), slag (26.64%), and SP (3.33%). The main effect plots are depicted in Figure 17 and Figure 18.

From the main effect plot for the mini slump, it can be concluded that there is a sharp increase in the slump when the water content increases. SP exhibits an almost identical trend.

Using the *S*/*N* ratio, each response is optimized. In this study, we used different objective functions for individual responses, such as maximization of the compressive strength, cylinder strengths for fine and coarse aggregates, and mini slump spread, and minimization of the temperature, pH, and penetration depth for fine and coarse aggregates samples. Therefore, the larger the better criterion is used for compressive strength, cylinder strengths for fine and coarse aggregates, and the mini slump spread, by using Equation (2). In contrast, the smaller-the-better criterion is used for temperature, pH, and penetration depth for fine and coarse aggregates samples, by using Equation (2).

Using these equations for the average *S*/*N* ratios of each response, optimal levels are obtained, as listed in Table 8. Good quality characteristics are represented by higher *S*/*N* ratios.

In Table 8, the highest *S*/*N* ratio of −24.94 is obtained for the mini slump spread, which designates that Grout 7 exhibited the optimized values for the mini slump spread. The ratio of −23.84 is obtained for the initial setting time, which specifies that Grout 5 had the optimized value for the initial setting time. Values of 37.41 and 13.48 for the UCS and bleeding, respectively, showed that Grout 1 produced the most optimized results. A bleeding value of 39.08 displays that Grout 6 exhibited the optimized value. With regard to temperature and water penetration depth, Grouts 9 and 3 showed the optimized results. For the cylinders with coarse and fine aggregates, Grouts 3 and 4 exhibited optimized levels, respectively.

All the steps involved in the multi-response optimization are depicted in Figure 14. The *S*/*N* ratios are normalized using Equations (4) and (5) and listed in Table 8 and Table 9. The Grey relational coefficient, listed in Table 10, was calculated using Equation (5). The response obtained via the GRC is listed in Table 11, which also shows the optimized levels of each factor.

In Table 11, the optimized levels for the best grout are listed, where water level 1, SP level 3, VMA level 2, and level 3 for slag are the optimized levels. The mix design ratios are listed in Table 12. The results for the confirmatory optimal grout mix are listed in Table 13.

## 5. Conclusions

In this study, laboratory experiments were performed to design a grout mix suitable for sealing underground water-filled karst cavities. The use of non-dispersible grout for filling karst cavities underwater is recommended to prevent grout washout under the flow of water. Many grout mixes, considering the target permeability of jointed rocks and open-jointed rocks, were tested, and the grout mix design suitable for implementation in actual environments was determined.

The key conclusions of this study on the optimal grout mix design can be summarized as follows:(1)This wide-ranging DOE study designed a new type of cement-based anti-washout grout. Laboratory tests were performed to study the effects of various constituents on mini slump spread, permeability, compressive strength, pH, dispersion, and temperature of the cement grout. The Grey relational analysis and Taguchi design of the joint experiments afforded a simple and cost-effective method for performing multiparametric optimization of cement grouts.(2)The main constituents of the proposed anti-washout grout as given in Table 12: (GRG Optimal Mix Design) are water, Portland cement, methylbenzyl cellulose, superplasticizer, and slag. The optimized proportions of these constituents for the best mini slump spread, yield stress, pH, dispersion, temperature, and permeability were as follows: 1% portion cement, 0.3% portion water, 0.1% portion slag, 0.03% portion SP, and 0.002% portion VMA. The optimized grout formula enabled strong anti-washout capacity, without dispersion, and a good setting time. Owing to the good fluidity of this mix using a low-pressure equipment, it can be used near and for deep subsurface targets.(3)Confirmatory experiments were performed to validate the results, which directed good agreement with the optimal grout mix. The optimized levels of mix constituents significantly increased the strength of the grout against water dilution, dispersion, segregation, and external bleeding. The addition of SP and methylbenzyl cellulose reduced the demand for water and helped to achieve a higher strength of 27 MPa, in addition to a low viscosity.(4)The results evidenced that Grey relational analysis is suitable for multi-objective optimization problems. Multi-objective ANOVA could obtain comparatively unimportant factors (*p* > 0.05) with respect to the prominent output responses. Some factors were insignificant for the rheological properties of the cement-based grout. Nevertheless, these insignificant factors can help adjust supplementary performances based on specific applications.(5)By determining the target permeability, as suggested in this research, researchers can optimize workability of grouts considering permeability and various hydraulic gradients in lab environments before using the grouts in the field.(6)The laboratory trials of the non-dispersive grout mixes presented in this paper were aimed to achieve an enhanced stable, non-dispersive and non-erodible grout mix with satisfactory workability and compressive strength properties. Site specific trials may be needed for further fine tuning of the proposed mix design, which is considered suitable for use in underwater construction, filling of partially filled cavities and open voids, and treatment of jointed rock.

## Figures and Tables

**Figure 1 materials-14-01587-f001:**
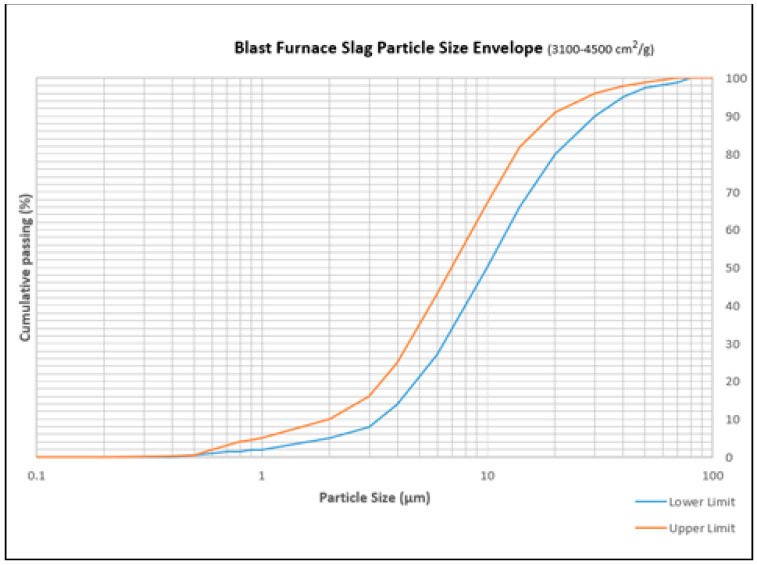
Blast furnace slag (BSF) cumulative particle size distribution (3100–4500 cm^2^/g).

**Figure 2 materials-14-01587-f002:**
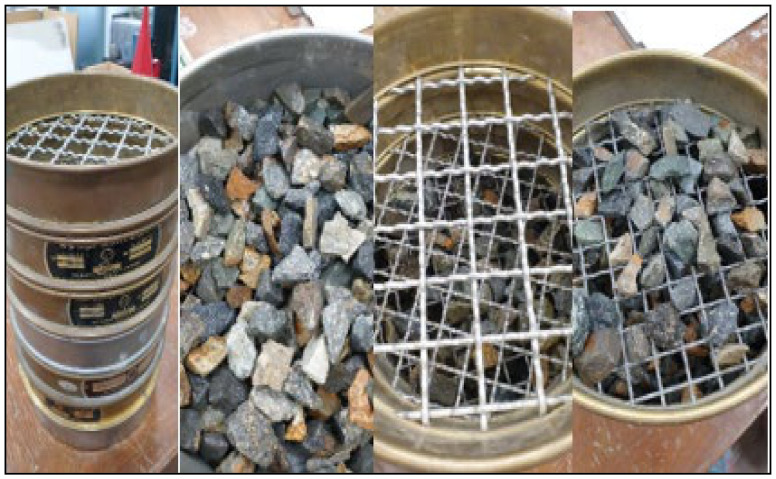
Sieve analysis for the aggregates.

**Figure 3 materials-14-01587-f003:**
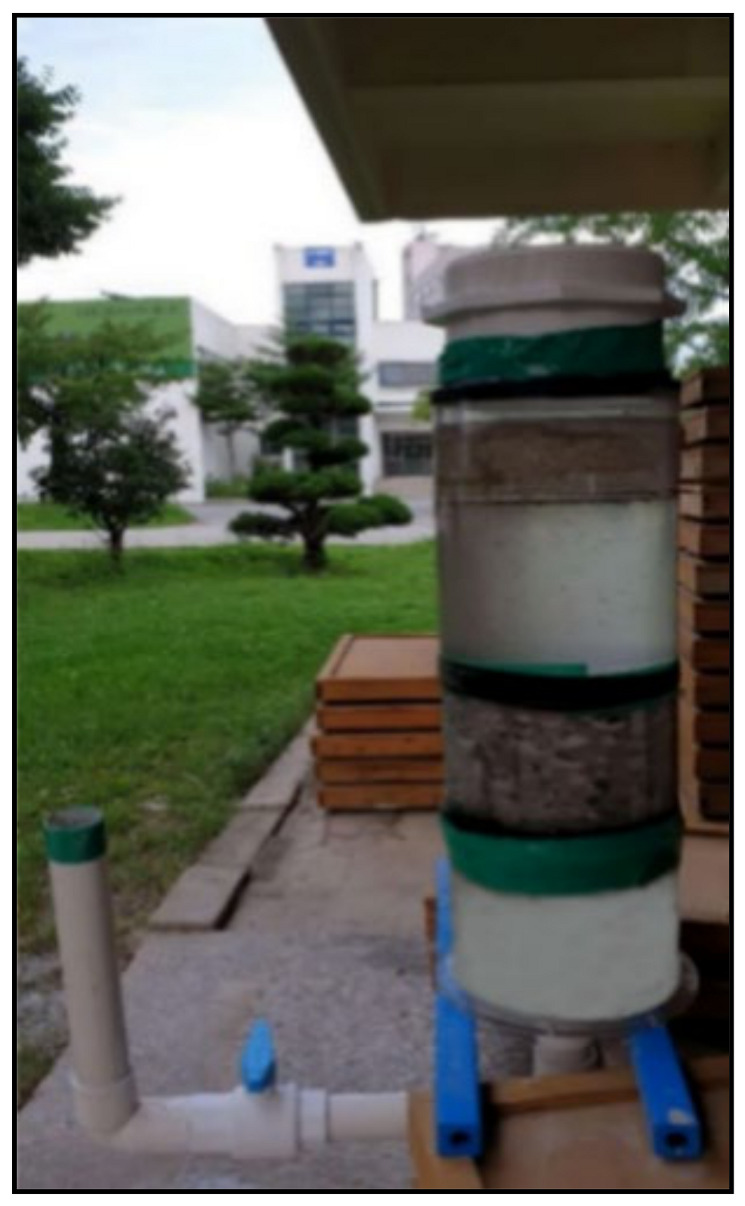
Falling head permeability test equipment.

**Figure 4 materials-14-01587-f004:**
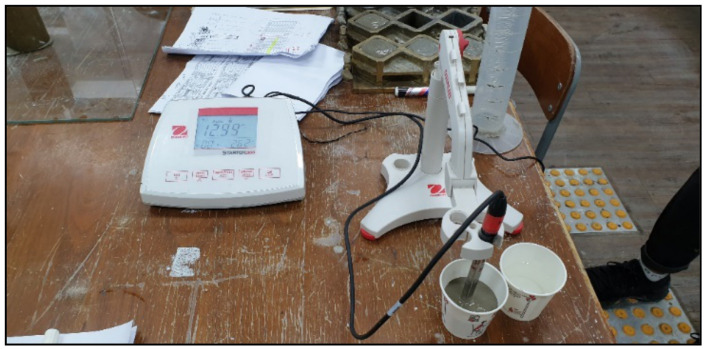
OHAUS starter3100 for measuring pH and temperature.

**Figure 5 materials-14-01587-f005:**
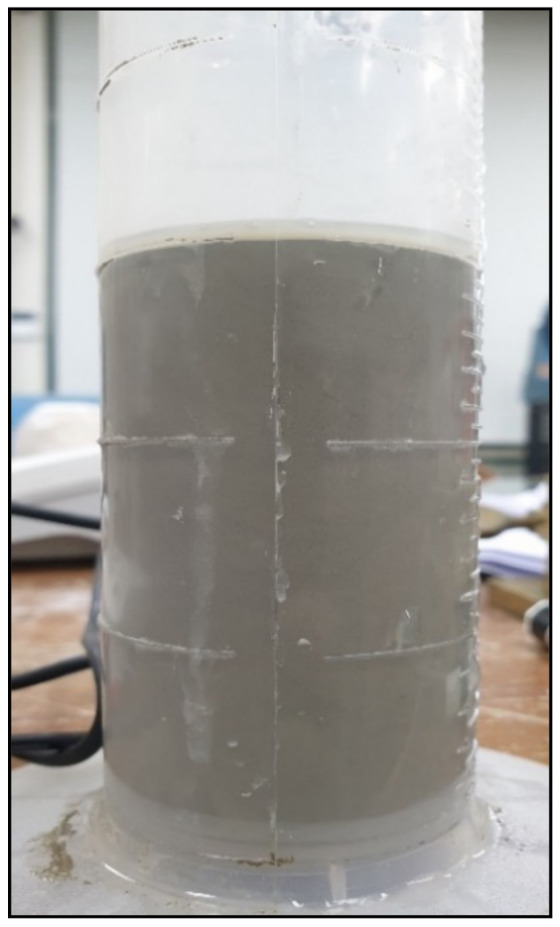
Method to test the bleeding of grout sample.

**Figure 6 materials-14-01587-f006:**
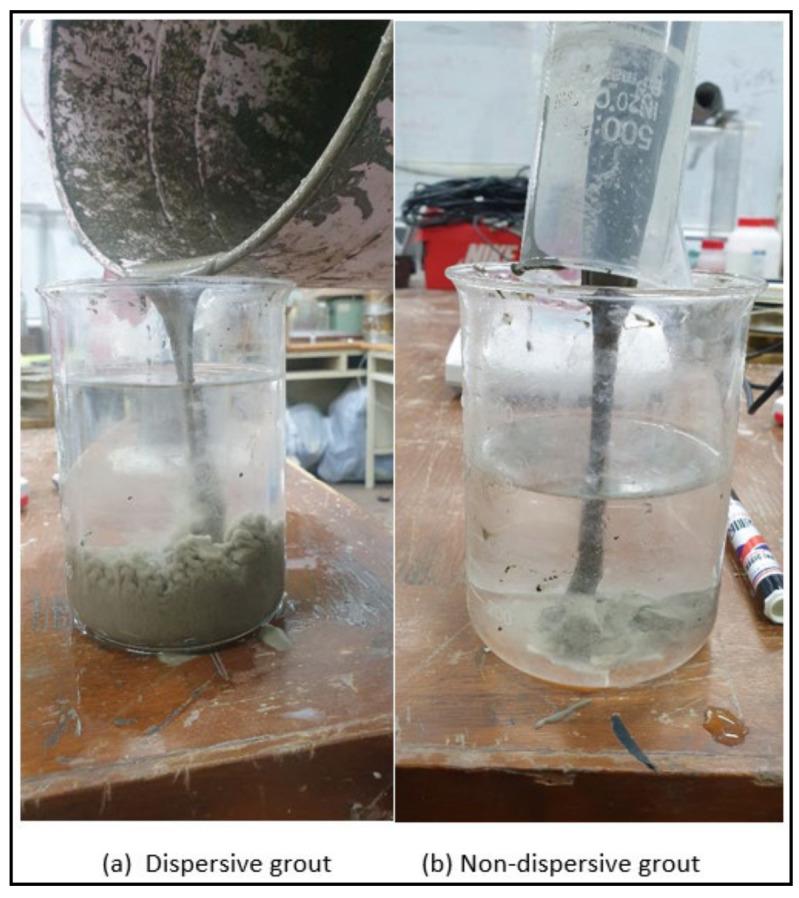
(**a**) Dispersive and (**b**) non-dispersive grouts testing.

**Figure 7 materials-14-01587-f007:**
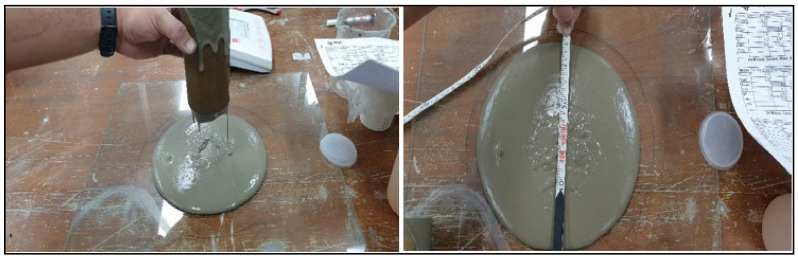
Mini slump spread measurement.

**Figure 8 materials-14-01587-f008:**
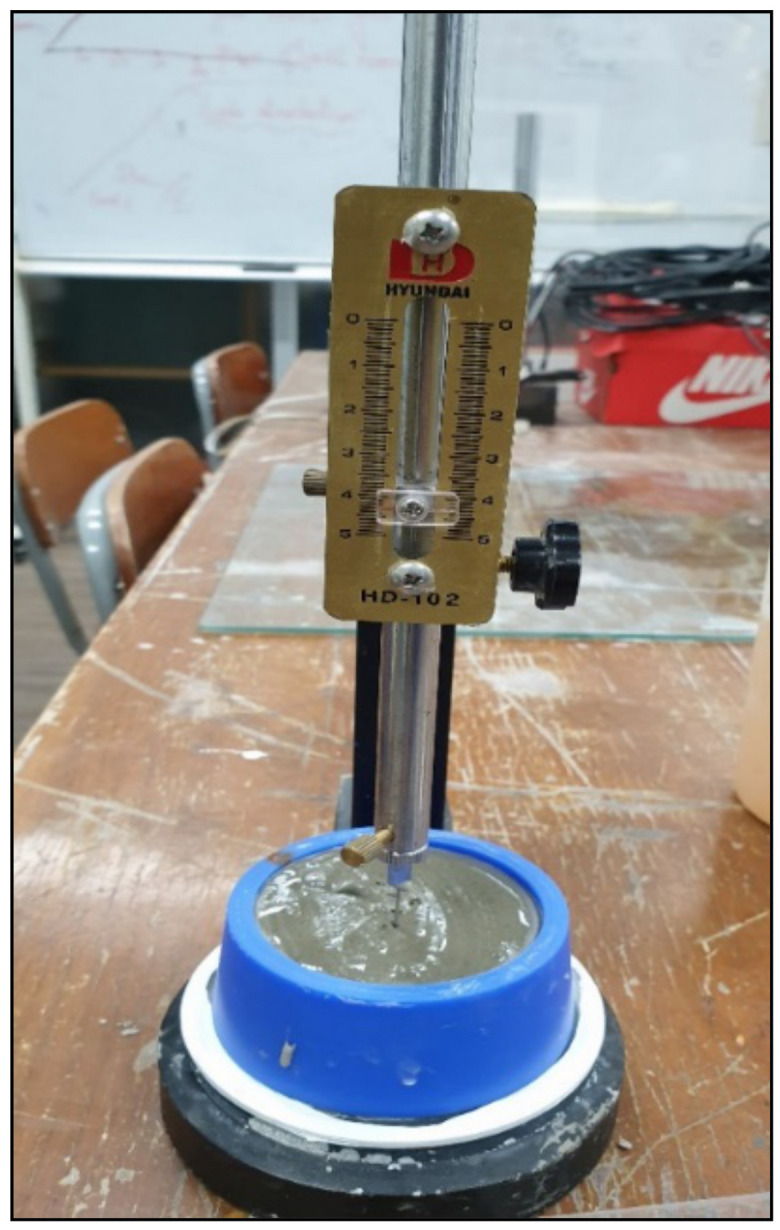
Vicat apparatus for measuring initial setting time.

**Figure 9 materials-14-01587-f009:**
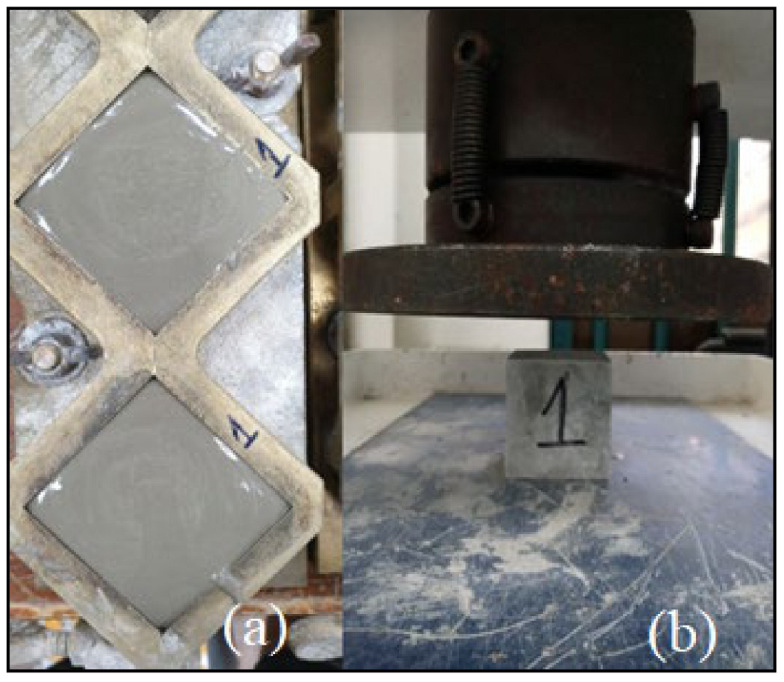
(**a**) Grout casting in Bronze mold and (**b**) compression test.

**Figure 10 materials-14-01587-f010:**
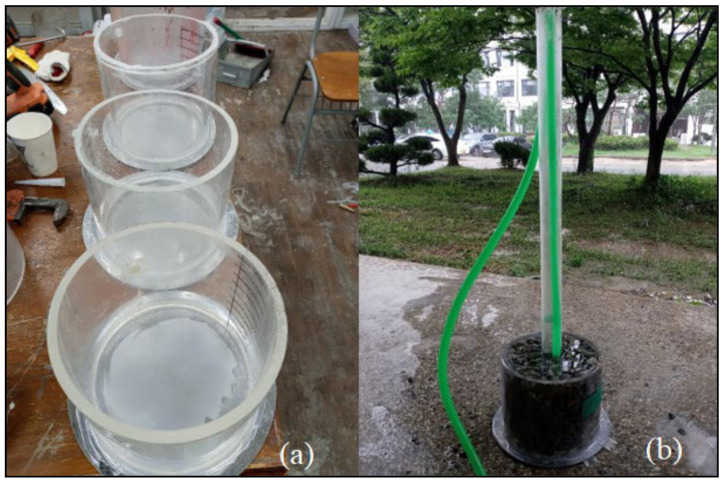
(**a**) Molds and (**b**) aggregate and water-filled mold.

**Figure 11 materials-14-01587-f011:**
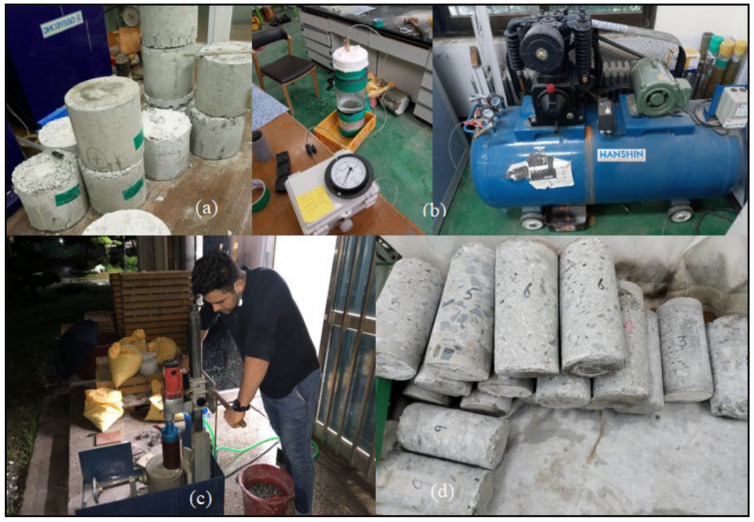
(**a**) Samples for the mass strength, (**b**) test for permeability, (**c**) use of rotary drill to extract cylinders, and (**d**) extracted cylinders having 100 mm diameter and 200 mm length.

**Figure 12 materials-14-01587-f012:**
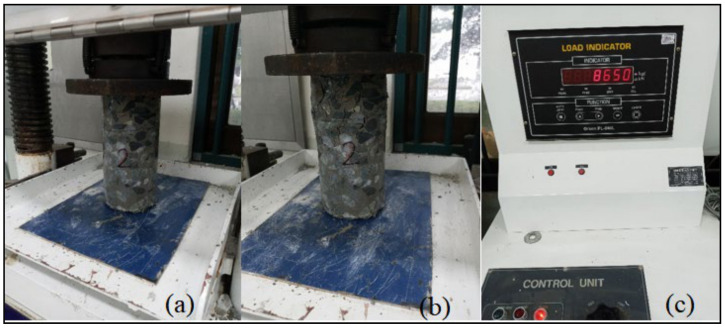
(**a**) Sample under compression, (**b**) failed sample, and (**c**) load indicator.

**Figure 13 materials-14-01587-f013:**
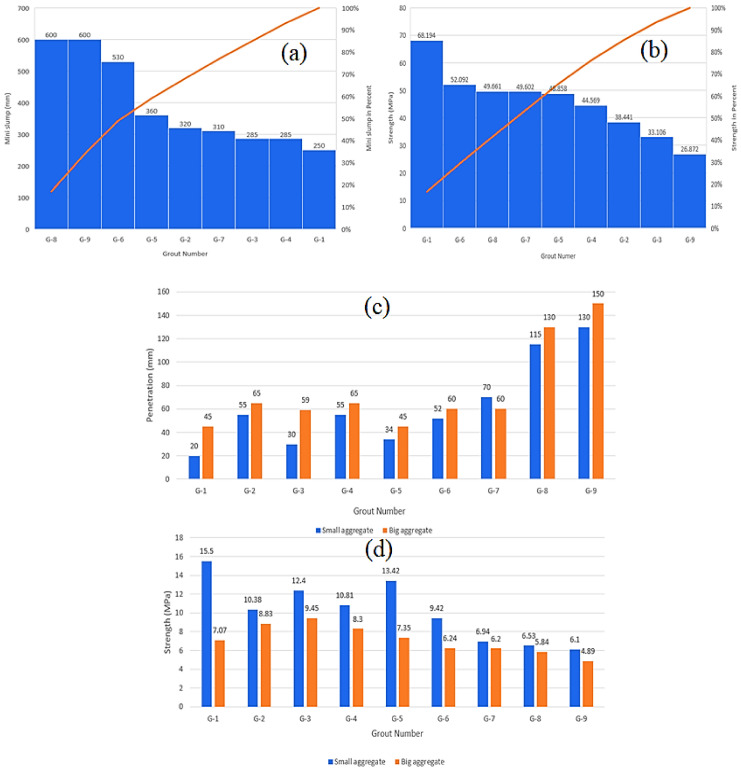
(**a**) Mini slump spread and (**b**) compressive strengths (MPa) of nine different grouts. (**c**) Mass permeability, water penetration depths, and (**d**) mass strengths of grout samples with fine and coarse aggregate.

**Figure 14 materials-14-01587-f014:**
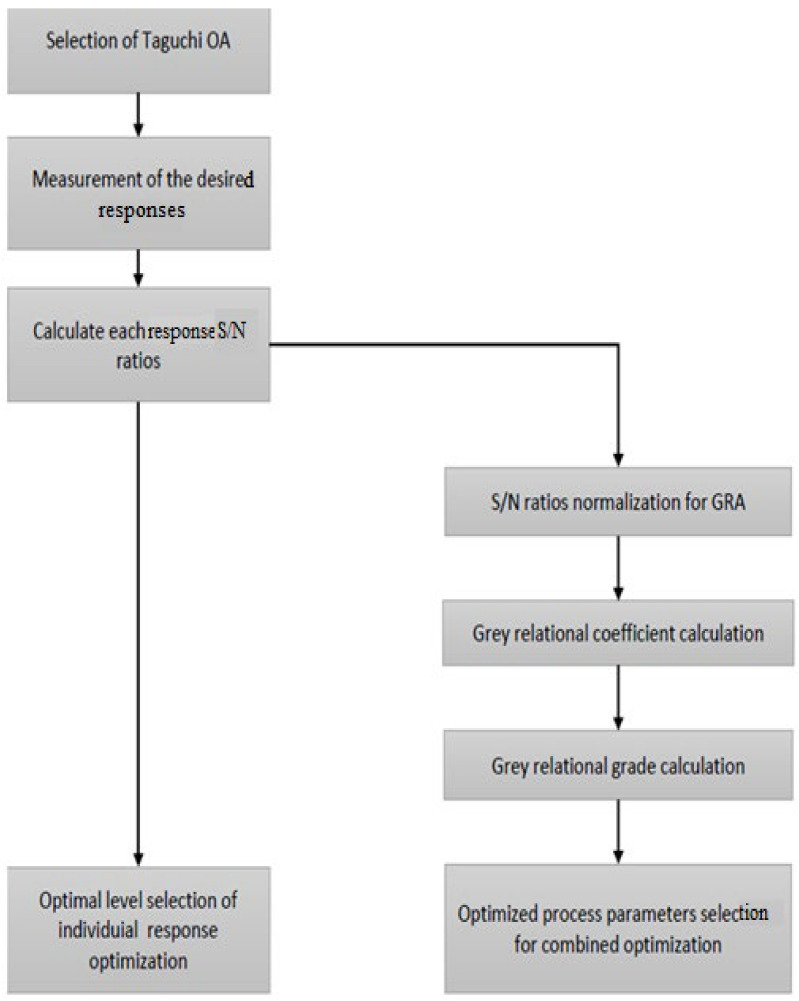
Concept of GRA-PCA.

**Figure 15 materials-14-01587-f015:**
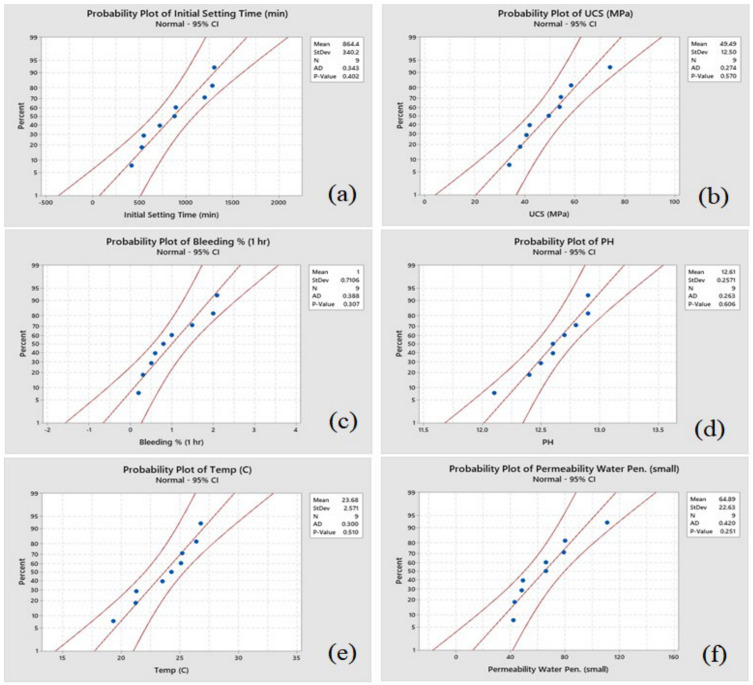
Normal probability plot response for (**a**) initial setting time, (**b**) UCS of cubes (MPa), (**c**) bleeding %, (**d**) pH, (**e**) temperature (°C), and (**f**) permeability water penetration for fine aggregate sample.

**Figure 16 materials-14-01587-f016:**
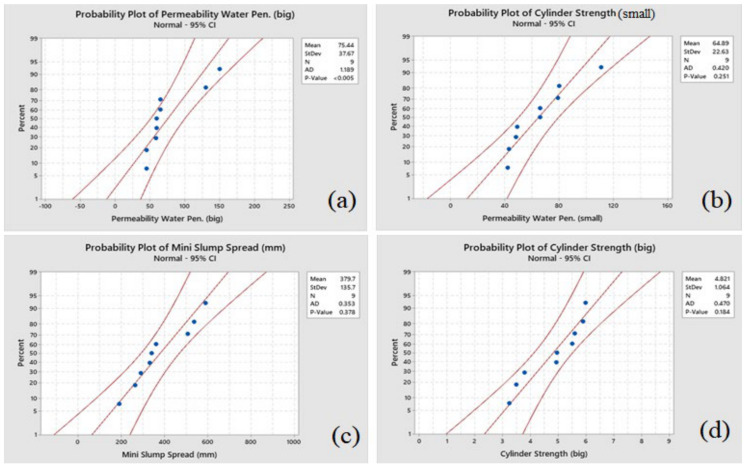
Normal probability plot for (**a**) permeability water penetration depth for coarse aggregates sample, (**b**) mass-cylinder strength with fine aggregates, (**c**) mini slump spread (mm), and (**d**) mass-cylinder strength with coarse aggregate.

**Figure 17 materials-14-01587-f017:**
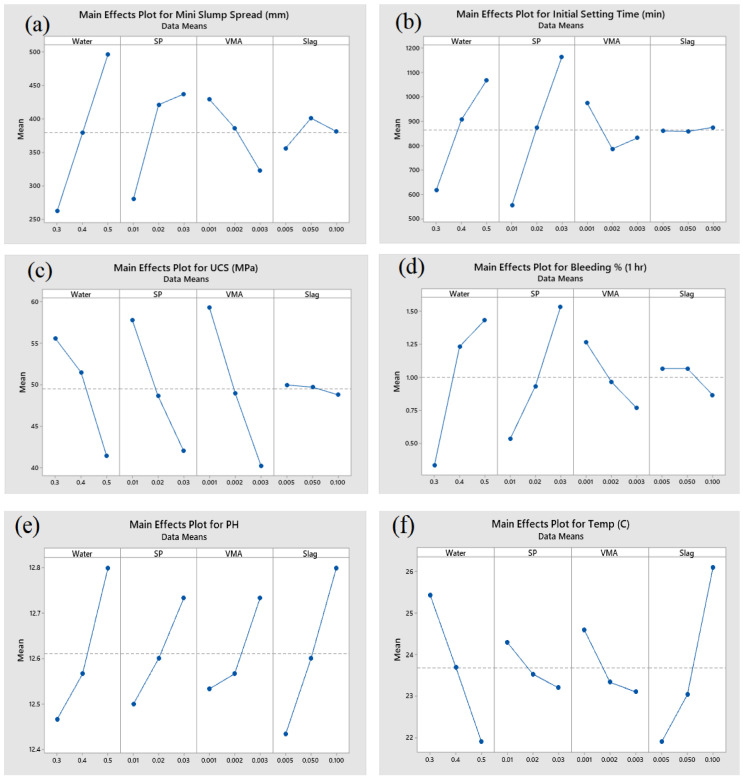
Main effect of plots for (**a**) mini slump spread, (**b**) initial setting time, (**c**) UCS of cubes (MPa), (**d**) bleeding %, (**e**) pH, and (**f**) temperature.

**Figure 18 materials-14-01587-f018:**
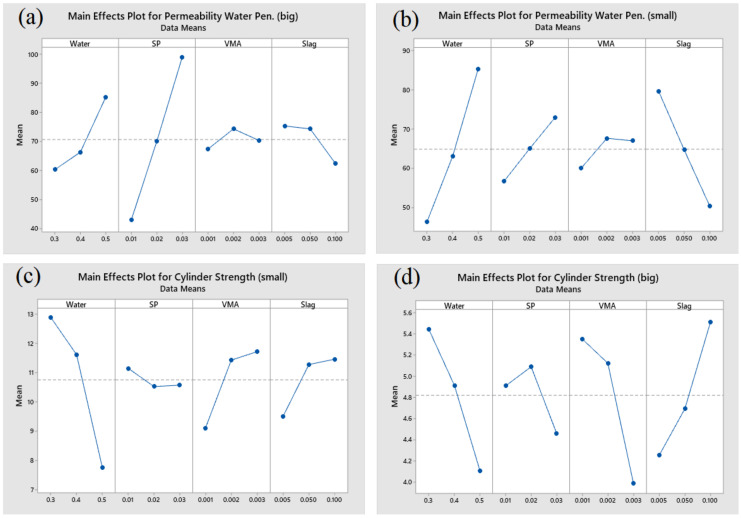
Main effect of plots for (**a**) permeability water penetration depth for coarse aggregate sample, (**b**) permeability water penetration depth for fine aggregate sample, (**c**) mass-cylinder strength with fine aggregate sample, and (**d**) mass-cylinder strength with coarse aggregate sample.

**Table 1 materials-14-01587-t001:** Chemical composition of BSF.

	Chemical Composition (%)										
Material	SiO_2_	Al_2_O_3_	Fe_2_O_3_	CaO	MgO	SO_3_	K_2_O	TiO_2_	P_2_O_5_	Total (%)	Specific Gravity
Blast furnace slag	34.76	14.50	0.48	41.71	6.87	0.13	0.44	0.62	0.03	99.54	2.81

**Table 2 materials-14-01587-t002:** Sieve analysis of coarse aggregate (16 mm).

Is Sieve Size	Weight Retained (g)	Cumulative Weight Retained (g)	Cumulative wt% Retained	Cumulative wt% Passing
40 mm	0	0	0	100
20 mm	0	0	0	100
16 mm	835	835	83.5	16.5
12.5 mm	80	915	91.5	8.5
10 mm	65	980	98	2
4.75 mm	20	1000	100	0
Pan	-	-	-	0
Total	1000	-	373	-
Hence Fineness Modulus (F.M.) = 3.68				

**Table 3 materials-14-01587-t003:** Sieve analysis of fine aggregate (10 mm).

Is Sieve Size	Weight Retained (g)	Cumulative Weight Retained (g)	Cumulative wt% Retained	Cumulative wt% Passing
40 mm	0	0	0	100
20 mm	0	0	0	100
16 mm	10	10	1	99
12.5 mm	90	100	10	90
10 mm	865	965	96.5	3.5
4.75 mm	35	1000	100	0
Pan	-	-	-	0
Total	1000	-	207.5	-
Hence Fineness Modulus (F.M.) = 2.075				

**Table 4 materials-14-01587-t004:** Ratios of nine different grouts Taguchi orthogonal array L9 ^(4)^.

Mixture	Cement	Water	Slag	SP	VMA
G-1	1	0.3	0.005	0.01	0.001
G-2	1	0.3	0.05	0.02	0.002
G-3	1	0.3	0.1	0.03	0.003
G-4	1	0.4	0.1	0.01	0.002
G-5	1	0.4	0.005	0.02	0.003
G-6	1	0.4	0.05	0.03	0.001
G-7	1	0.5	0.05	0.01	0.003
G-8	1	0.5	0.1	0.02	0.001
G-9	1	0.5	0.005	0.03	0.002

^(4)^ The L9 orthogonal array is meant for understanding the effect of 4 independent factors each having 3 factor level values. This array assumes that there is no interaction between any two factors.

**Table 5 materials-14-01587-t005:** Mix proportions of the grouting materials.

**1st Mixture: Cement, Water, Slag, and VMA**					**2nd Mixture: Cement, Water, Slag, and VMA**				
**Mixture**	**Density (g/cm^3^)**	**Proportion**	**Weight (g)**	**Volume (cm^3^)**	**Mixture**	**Density (g/cm^3^)**	**Proportion**	**Weight (g)**	**Volume (cm^3^)**
Cement	3.15	1	7154.45	2271.25	Cement	3.15	1	6877.15	2183.22
Water	1	0.3	2146.33	2146.33	Water	1	0.3	2063.14	2063.14
Slag	2.9	0.005	35.77	12.34	Slag	2.9	0.05	343.86	118.57
SP	1.2	0.01	71.54	59.62	SP	1.2	0.02	137.54	114.62
VMA	1.2	0.001	7.15	5.96	VMA	1.2	0.002	13.75	11.46
Total	-	-	9415	4500	Total	-	-	9435	4500
				4496					4491
**3rd Mixture: Cement, Water, Slag, and VMA**					**4th Mixture: Cement, Water, Slag, and VMA**				
**Mixture**	**Density (g/cm^3^)**	**Proportion**	**Weight (g)**	**Volume (cm^3^)**	**Mixture**	**Density (g/cm^3^)**	**Proportion**	**Weight (g)**	**Volume (cm^3^)**
Cement	3.15	1	6603.26	2096.27	Cement	3.15	1	5894.16	1871.16
Water	1	0.3	1980.98	1980.98	Water	1	0.4	2357.67	2357.67
Slag	2.9	0.1	660.33	227.70	Slag	2.9	0.1	589.42	203.25
SP	1.2	0.03	198.10	165.08	SP	1.2	0.01	58.94	49.12
VMA	1.2	0.003	19.81	16.51	VMA	1.2	0.002	11.79	9.82
Total	-	-	9462	4500	Total	-	-	8912	4500
				4487					4491
**5th Mixture: Cement, Water, Slag, and VMA**					**6th Mixture: Cement, Water, Slag, and VMA**				
**Mixture**	**Density (g/cm^3^)**	**Proportion**	**Weight (g)**	**Volume (cm^3^)**	**Mixture**	**Density (g/cm^3^)**	**Proportion**	**Weight (g)**	**Volume (cm^3^)**
Cement	3.15	1	6076.43	1929.03	Cement	3.15	1	5910.98	1876.50
Water	1	0.4	2430.57	2430.57	Water	1	0.4	2364.39	2364.39
Slag	2.9	0.005	30.38	10.48	Slag	2.9	0.05	295.55	101.91
SP	1.2	0.02	121.53	101.27	SP	1.2	0.03	177.33	147.77
VMA	1.2	0.003	18.23	15.19	VMA	1.2	0.001	5.91	4.93
Total	-	-	8677	4500	Total	-	-	8754	4500
				4487					4496
**7th Mixture: Cement, Water, Slag, and VMA**					**8th Mixture: Cement, Water, Slag, and VMA**				
**Mixture**	**Density (g/cm^3^)**	**Proportion**	**Weight (g)**	**Volume (cm^3^)**	**Mixture**	**Density (g/cm^3^)**	**Proportion**	**Weight (g)**	**Volume (cm^3^)**
Cement	3.15	1	5306.16	1684.49	Cement	3.15	1	5170.56	1641.45
Water	1	0.5	2653.08	2653.08	Water	1	0.5	2585.28	2585.28
Slag	2.9	0.05	265.31	91.49	Slag	2.9	0.1	517.06	178.30
SP	1.2	0.01	53.06	44.22	SP	1.2	0.02	103.41	86.18
VMA	1.2	0.003	15.92	13.27	VMA	1.2	0.001	5.17	4.31
Total	-	-	8294	4500	Total	-	-	8381	4500
				4487					4496
**9th Mixture: Cement, Water, Slag, and VMA**									
**Mixture**	**Density (g/cm^3^)**	**Proportion**	**Weight (g)**	**Volume (cm^3^)**					
Cement	3.15	1	5309.47	1685.54					
Water	1	0.5	2654.73	2654.73					
Slag	2.9	0.005	26.55	9.15					
SP	1.2	0.03	159.28	132.74					
VMA	1.2	0.002	10.62	8.85					
Total	-	-	8161	4500					
				4491					

**Table 6 materials-14-01587-t006:** Ratios of various constituents based on the Taguchi array.

Water	SP	VMA	Slag		
0.3	0.01	0.001	0.005		
0.4	0.02	0.002	0.05		
0.5	0.03	0.003	0.1		
**Mixture**	**Cement**	**Water**	**Slag**	**SP**	**VMA**
G-1	1	0.3	0.005	0.01	0.001
G-2	1	0.3	0.05	0.02	0.002
G-3	1	0.3	0.1	0.03	0.003
G-4	1	0.4	0.1	0.01	0.002
G-5	1	0.4	0.005	0.02	0.003
G-6	1	0.4	0.05	0.03	0.001
G-7	1	0.5	0.05	0.01	0.003
G-8	1	0.5	0.1	0.02	0.001
G-9	1	0.5	0.005	0.03	0.002
**Mixture**	**Initial Setting Time** **(h:min)**	**Dispersion**	**Bleeding %** **(1 h)**	**pH**	**Temperature (°C)**
G-1	7:30	No	0.3	12.2	26.2
G-2	10:15	No	0.4	12.9	24.9
G-3	15:00	No	0.3	12.9	24.8
G-4	9:20	No	0.3	12.6	24.4
G-5	15:30	No	0.1	12.3	23.4
G-6	23:20	Yes	2	12.5	24
G-7	6:00	No	0.5	12.5	23.3
G-8	20:30	Yes	3	12.8	23.1
G-9	23:00	Yes	3	12.7	23.3

**Table 7 materials-14-01587-t007:** ANOVA results for mini slump, pH, bleeding, UCS of cubes, initial setting time, and compressive strength for the cylinders of coarse and fine aggregates.

Response	Source	DoF	Adj SS	Adj MS	F-Value	*p*-Value	% Contribution
Mini Slump Spread (mm)	Water	1	82,134	82,134	32.12	0.005	55.74
	SP	1	36,974	36,973.5	14.46	0.019	25.09
	VMA	1	17,174	17,173.5	6.72	0.061	11.65
	Slag	1	853	853.1	0.33	0.595	0.58
	Error	4	10,230	2557.5	-	-	6.94
	Total	8	147,364	-	-	-	100.00
Initial Setting Time (min)	Water	1	303,750	303,750	33.84	0.004	32.81
	SP	1	555,104	555,104	61.85	0.001	59.95
	VMA	1	30,817	30,817	3.43	0.138	3.33
	Slag	1	352	352	0.04	0.853	0.04
	Error	4	35,900	8975	-	-	3.88
	Total	8	925,922	-	-	-	100.00
UCS (MPa)	Water	1	302.55	302.545	52.85	0.002	24.22
	SP	1	372.8	372.803	65.12	0.001	29.84
	VMA	1	548.76	548.763	95.85	0.001	43.93
	Slag	1	2.19	2.186	0.38	0.57	0.18
	Error	4	22.9	5.725	-	-	1.83
	Total	8	1249.2	-	-	-	100.00
Bleeding % (1 h)	Water	1	1.815	1.815	25.21	0.007	44.93
	SP	1	1.5	1.5	20.84	0.01	37.13
	VMA	1	0.375	0.375	5.21	0.085	9.28
	Slag	1	0.06207	0.06207	0.86	0.406	1.54
	Error	4	0.28793	0.07198	-	-	7.13
	Total	8	4.04	-	-	-	100.00
pH	Water	1	0.16667	0.166667	36.17	0.004	31.51
	SP	1	0.08167	0.081667	17.72	0.014	15.44
	VMA	1	0.06	0.06	13.02	0.023	11.34
	Slag	1	0.20212	0.202124	43.86	0.003	38.22
	Error	4	0.01843	0.004608	-	-	3.48
	Total	8	0.52889	-	-	-	100.00
Temp (°C)	Water	1	18.727	18.7267	35.76	0.004	35.42
	SP	1	1.815	1.815	3.47	0.136	3.43
	VMA	1	3.375	3.375	6.45	0.064	6.38
	Slag	1	26.864	26.8643	51.3	0.002	50.81
	Error	4	2.095	0.5237	-	-	03.96
	Total	8	52.876	-	-	-	100.00
Water Penetration (fine)	Water	1	2281.5	2281.5	171.3	0.000	55.69
	SP	1	400.17	400.17	30.05	0.005	9.77
	VMA	1	73.5	73.5	5.52	0.079	1.79
	Slag	1	1288.45	1288.45	96.74	0.001	31.45
	Error	4	53.27	13.32	-	-	1.30
	Total	8	4096.89	-	-	-	100.00
Water Penetration (coarse)	Water	1	937.5	937.5	18.74	0.012	15.33
	SP	1	4704	4704	94.01	0.001	76.91
	VMA	1	13.5	13.5	0.27	0.631	0.22
	Slag	1	260.84	260.84	5.21	0.084	4.26
	Error	4	200.16	50.04	-	-	3.27
	Total	8	6116	-	-	-	100.00
Cylinder Strength (fine)	Water	1	2.7001	2.7001	13.05	0.023	62.91
	SP	1	0.3015	0.3015	1.46	0.294	0.79
	VMA	1	2.8085	2.8085	13.57	0.021	16.26
	Slag	1	2.4109	2.4109	11.65	0.027	8.80
	Error	4	0.8276	0.2069	-	-	11.24
	Total	8	9.0486	-	-	-	100
Cylinder Strength (coarse)	Water	1	2.7001	2.7001	13.05	0.023	29.84
	SP	1	0.3015	0.3015	1.46	0.294	3.33
	VMA	1	2.8085	2.8085	13.57	0.021	31.04
	Slag	1	2.4109	2.4109	11.65	0.027	26.64
	Error	4	0.8276	0.2069	-	-	9.15
	Total	8	9.0486	-	-	-	100.00

DoF: Degree of Freedom; Adj SS: Adjusted sums of squares; Adj MS: Adjusted Mean Squares.

**Table 8 materials-14-01587-t008:** S/N ratios of response variables.

Exp. No.	S/N Ratios of Response Variables									
	Mini Slump Spread (mm)	Initial Setting Time (min)	UCS (MPa)	Bleeding % (1 h)	pH	Temperature (°C)	Water Penetration (Coarse)	Water Penetration (Fine)	Cylinder Strength (Fine)	Cylinder Strength (Coarse)
1	−45.56	−53.05	37.41	13.98	5.83	−28.03	−33.62	−30.63	20.34	14.65
2	−33.38	−50.09	34.72	10.46	13.51	−27.71	−33.80	−36.52	22.85	15.40
3	−41.19	−29.70	31.62	6.02	10.79	−28.56	−32.46	−38.06	23.17	15.48
4	−39.15	−50.49	35.35	4.44	39.08	−28.43	−32.67	−30.63	22.55	15.56
5	−31.75	−23.84	32.44	0.00	19.08	−26.53	−38.06	−36.90	20.92	11.55
6	−42.23	−52.98	34.66	−6.44	39.08	−27.42	−36.39	−39.55	20.27	13.91
7	−24.94	−43.19	32.17	1.94	14.48	−26.57	−37.73	−35.71	19.68	10.18
8	−46.46	−50.52	33.92	−3.52	10.79	−27.99	−36.39	−37.27	16.36	14.96
9	−43.99	−52.48	30.60	−6.02	21.02	−25.71	−40.91	−41.73	16.90	10.83
cri	Nominal	Nominal	Larger	Smaller	Smaller	Smaller	Smaller	Smaller	Larger	Larger
**Min**	**−46.46**	**−53.05**	**30.60**	**−6.44**	**5.83**	**−28.56**	**−40.91**	**−41.73**	**16.36**	**10.18**
**Max**	**−24.94**	**−23.84**	**37.41**	**13.98**	**39.08**	**−25.71**	**−32.46**	**−30.63**	**23.17**	**15.56**

**Table 9 materials-14-01587-t009:** Grey relational analysis for data normalization.

Exp. No.	Normalization									
	Mini Slump Spread (mm)	Initial Setting Time (min)	UCS (MPa)	Bleeding %(1 h)	pH	Temperature (°C)	Permeability of Water Penetration (Fine)	Permeability of Water Penetration (Coarse)	Cylinder Strength (Fine)	Cylinder Strength (Coarse)
1	0.5640	0.9978	1.0000	0.0000	0.9233	0.8125	0.1374	0.0000	0.5845	0.8299
2	0.5877	0.9005	0.6052	0.1724	0.8281	0.7017	0.1586	0.5309	0.9530	0.9701
3	0.8684	0.2012	0.1502	0.3897	0.9163	1.0000	0.0000	0.6697	1.0000	0.9851
4	0.9895	0.9141	0.6984	0.4672	0.0000	0.9542	0.0242	0.0000	0.9091	1.0000
5	0.4739	0.0000	0.2700	0.6845	0.6476	0.2860	0.6630	0.5652	0.6700	0.2539
6	0.7956	0.9996	0.5970	1.0000	0.0000	0.5997	0.4651	0.8042	0.5746	0.6926
7	0.0000	0.6640	0.2311	0.5896	0.7968	0.3003	0.6237	0.4575	0.4876	0.0000
8	0.5014	0.9151	0.4880	0.8569	0.9163	0.8004	0.4651	0.5980	0.0000	0.8886
9	0.6732	0.9824	0.0000	0.9793	0.5848	0.0000	1.0000	1.0000	0.0790	0.1204
**Max**	**1**	**1**	**1**	**1**	**1**	**1**	**1**	**1**	**1**	**1**

**Table 10 materials-14-01587-t010:** Grey relational coefficients.

Exp. No.	Grey Relational Coefficients											
	Mini Slump Spread (mm)	Initial Setting Time (Min)	UCS (MPa)	Bleeding %(1 h)	pH	Temperature (°C)	Water Penetration (Fine)	Water Penetration (Coarse)	Cylinder Strength (Fine)	Cylinder Strength (Coarse)	GRG	Rank
1	0.5403	0.9965	1.0000	0.3333	1.0000	0.7273	0.3669	0.3333	0.5461	0.7461	0.6590	**3**
2	0.5544	0.8345	0.5588	0.3766	0.8401	0.6263	0.3727	0.5160	0.9141	0.9436	0.6537	**5**
3	0.8051	0.3851	0.3704	0.4503	0.9863	1.0000	0.3333	0.6022	1.0000	0.9711	0.6904	**1**
4	1.0000	0.8540	0.6238	0.4841	0.3513	0.9161	0.3388	0.3333	0.8461	1.0000	0.6748	**2**
5	0.4923	0.3334	0.4065	0.6131	0.6446	0.4119	0.5974	0.5349	0.6024	0.4013	0.5038	**8**
6	0.7206	1.0000	0.5537	1.0000	0.3513	0.5554	0.4831	0.7186	0.5403	0.6193	0.6542	**4**
7	0.3357	0.5984	0.3941	0.5492	0.7981	0.4168	0.5706	0.4796	0.499	0.3333	0.4970	**9**
8	0.5060	0.8555	0.4941	0.7775	0.9863	0.7147	0.4831	0.5544	0.333	0.8178	0.6523	**6**
9	0.6126	0.9667	0.3333	0.9602	0.5963	0.3333	1.0000	1.0000	0.359	0.3624	0.6517	**7**

**Table 11 materials-14-01587-t011:** Response table for average of GRG’s.

Level	Water	SP	VMA	Slag
1	**0.6677**	0.6102	0.6552	0.6048
2	0.6109	0.6033	**0.6601**	0.6016
3	0.6003	**0.6654**	0.5637	**0.6725**
Delta	0.06740	0.0622	0.0963	0.0708241
**Rank**	**3**	**4**	**1**	**2**

**Table 12 materials-14-01587-t012:** GRG optimal mix design.

Mixture	Density (g/cm^3^)	Proportion	Weight (g)	Volume (cm^3^)
Cement	3.15	1	2647.19	840.38
Water	1	0.3	794.16	794.16
Slag	2.9	0.1	264.72	91.28
SP	1.2	**0.03**	79.42	66.18
VMA	1.2	0.002	5.29	4.41
Total	-	-	3791	1800
				1796

**Table 13 materials-14-01587-t013:** Results for the confirmatory optimal grout.

Sample	Cube Strength (MPa) in 7 days	Cube Strength (MPa) in 28 day	Strength of Cylinder with Coarse Aggregates (7 days)	Strength of Cylinder with Fine Aggregates (7 days)	Water Penetration Depth Coarse Aggregates Sample (mm)	Water Penetration Depth Fine Aggregate Sample (mm)
Optimal mix	18.1	27.84	3.69	4.82	95	88
**PH**	**Bleeding (%)**	**Temp (°C)**	**Initial Setting Time** **(h)**	**Mini Slump Spread** **(mm)**	**Dispersion**	
12.08	2.4	22.7	8	550	No	

## Data Availability

The data presented in this study are available on request from the corresponding author.

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
