# Peer review of "Non-Dispersive Anti-Washout Grout Design Based on Geotechnical Experimentation for Application in Subsidence-Prone Underwater Karstic Formations"

_materials, 2021, doi:10.3390/ma14071587_

Round 1

Reviewer 1 Report

This manuscript presented extensive experimental work on a non-dispersive anti-washout grout. The grout consists of ordinary Portland cement, slag, superplasticizer and methylbenzyl cellulose. Taguchi orthogonal array was used for designing and optimizing the mix design of grout. Influence of raw materials on properties, e.g., slump, pH, grout dispersion, bleeding, setting time, compressive strength and permeability of grouts, was evaluated. However, the scientific/engineering problem of this manuscript was not defined, even though a comprehensive literature review was carried out on existing grouts. Moreover, the motivation of this manuscript was unclear. Scientific significance of this manuscript needs to be improved. Therefore, publication of this manuscript in "materials" is not recommended. 

Reviewer 2 Report

  • The article lacks a more detailed description of the slag. This is mainly a grain sieve analysis and possibly a chemical analysis
  • In the Table 8. is the Error without any explanation
  • the overall description of Table 8 needs to be better clarified
  • The individual points in the conclusion should be better described to the reader. Some are unnecessarily too complicated. 

Reviewer 3 Report

A. THE STRENGTHS OF THE PAPER

  • Application of the experiment planning method

B. THE WEAKNESSES OF THE PAPER

  • “REFERENCES" (LINE 456-492) needs ordering

Please, ORDER and COMPLETE the text:

  • enter authors and titles of the publication correctly (surname / first letter of the first name)
  • In the title of a paper use ‘capital letter’ only at the beginning of the sentence (in the first word)!!!, e.g. Laboratory Evaluation of a Unique Anti-Washout Admixture in Grouts / Laboratory evaluation of a unique anti-washout admixture in grouts
  • correct punctuation errors
  • enter "pp." or give page numbers without "pp.", etc.

For example:

LINE (459) University of Achen Aachen

LINE (460) Quinziéme Congres des Grandes Barrage Grands Barrages

LINE (462) Eklund, H.; Stille, Eklund D., Stille H.

LINE (462-463, 465) Tunneling and Underground Space Technology, Elsevier

LINE (464) Shucai Li.; Song S. et al.

LINE (466) Bury, J.R.; and H. Farzam Bury J.R., Farzam H. (…) Symposium , ACI SP-173,

LINE (468-469) Kim Uk-Gie. Kim U.-G. et al. (…) 2013, (…) , 44-55 25-33

LINE (470-471) Cui, W.; Cui W. et al. (…) Grey grey (…) SCOPUS Construction and Building Materials

LINE (472-473) Johann Plank Lang A., Plank J. (…) 2015, 132, 37 … ( ? )

LINE (474) Tawara H. et al.; underwater Underwater anti-washout non-shrink grout

LINE (476) ) Windal Scott Bray. Bray W.S., Brandl A. (…) methylhydroxyethyl cellulose

LINE (477) Sahara H.; Concrete composition ( ? ) JB ( ? )

LINE (486) SidharthaPanda Panda S. (…) Elsevier International Journal of Electrical Power & Energy Systems

  • LINGUISTIC MISTAKES
  1. Inappropriate use of “English Tenses” (Past/Present/Future) (SEQUENCE of Tenses)
  2. Lack of synonyms; constant use of one word: Li developed (LINE 12)/ Jeff developed (LINE 54)/ Cui developed (LINE 57), Jeff developed (LINE (12), Jeff developed (LINE 57), Jeff developed (LINE (12)instead of using synonyms

Please, READ the whole text carefully and CORRECT linguistic mistakes !!!

The EXAMPLES !!! of linguistic mistakes:

LINE (12) super plasticizer superplasticizer

LINE (12-13) A series of laboratory experiments is were performed → ENGLISH TENSE

LINE (237-238) (C) (a) Mini slump spread and (b) compressive strenths strength (MPa) (…) (c) Mass permeability, water penetration depths depth, (d) (…) strengths strength of grout samples with fine and coarse aggregates

LINE (307) with fine aggregate with fine aggregates

LINE (308) with coarse aggregate with coarse aggregates

LINE (375) the larger-the-better criterion → “the larger the better” criterion

LINE (382-387) In Table 9, the highest S/N ratio of -24.94 is obtained for the mini slump spread, which indicates that Grout 7 exhibited the optimized values ( ? ) for the mini slump spread. The ratio of -23.84 is obtained for the initial setting time, which indicates that Grout 5 had ( ? ) the optimized value for the initial setting time. Values of 37.41 and 13.48 for the UCS and bleeding, respectively, indicated that Grout 1 produced the most optimized results. A bleeding value of 39.08 indicates that Grout 6 exhibited the optimized value. → CONSTANT USE OF ONE PHRASE/WORD: “is obtained” / “indicate”; ENGLISH TENSES !!!

  • OTHER ERRORS/ UNCLEAR VALUES PRESENTED

LINE (23-24) water level 1 (0.3%), SP level 3 (0.01%), methylbenzyl cellulose level 2 (0.002%), and slag level 3 (0.1%) - PERCENT of what ( ?) (by mass/ by volume - ?)

LINE (224) 1% portion cement, 0.3% portion water, 0.1% portion slag, 0.03% portion SP, and 0.002% portion VMA - PERCENT of what ( ?) (by mass/ by volume - ?)

LINE (144) ph pH

LINE (159, 169) the V-cat apparatus → Vicat (!) apparatus

Round 2

Reviewer 1 Report

The manuscript has been improved. There is still one comment: please avoid repeating to present experimental results in tables if they have already been shown in figures.

Author Response

According to the suggested point the modifications are made as follows:

From Table 6, the section for the mini-slump spread results is removed as it was present in Figure 13 and as well as Table 7 and 8 are completely deleted as Figure 13 contains the histograms of these results. Table numbers are updated again due to the deletion of Tables 7 and 8.
